# Production Choices and Food Security: A Review of Studies Based on a Micro-Diversity Perspective

**DOI:** 10.3390/foods13050771

**Published:** 2024-03-01

**Authors:** Yanfang Huang, Yuying Yang, Fengying Nie, Xiangping Jia

**Affiliations:** 1Institute of Agricultural Information, Chinese Academy of Agricultural Sciences, Beijing 100081, China; huangyanfang01@caas.cn (Y.H.); niefengying@caas.cn (F.N.); 2School of Agricultural Economics and Rural Development, Renmin University of China, Beijing 100872, China; yangyuying@ruc.edu.cn

**Keywords:** dietary diversity, agricultural production diversity, bibliometrix, smallholder households

## Abstract

Given the ‘subsistence’ character of smallholder production, agricultural production diversification is often seen as an effective strategy for smallholders to improve their diets’ diversity and nutritional status, yet the existing evidence remains inadequate. The study applies bibliometric data from the “Web of Science” database to synthesize 46 papers from developing countries to explore the relationship between production diversity, dietary diversity, and nutrition in smallholder households. The study identifies the most influential journals, authors, organizations, and countries and reveals research themes related to agricultural production and food security. This data analysis can help researchers target potential collaborators and access influential literature in agricultural production diversity and dietary diversity research. In addition, the results showed that agricultural production diversity potentially influences households’ dietary diversity, with mixed results: Agricultural production diversification is the primary way to improve food and nutritional security among smallholder families with low socio-economic status, inaccessible transportation, and poverty; market access and trade have more potential to improve dietary diversity among smallholder households with well-developed markets and higher income levels; the significant measures of agricultural production diversity include Crop Counts, FGPD, SI, and SWDI; the significant measures of dietary diversity include HDDS and IDDS. This paper provides a roadmap for agricultural production and food security researchers by conducting a systematic review of the literature, summarizing some research methods and perspectives applicable to local socio-economic development.

## 1. Introduction

Eradicating hunger and malnutrition is integral to Goal 2 of the United Nations Sustainable Development Goals (SDGs). Over the past two decades, the role of increased agricultural productivity in reducing hunger has been self-evident [1]. However, malnutrition remains prevalent [2,3] and is mainly characterized by poor dietary quality and low levels of dietary diversity [4]. In China, the “all-encompassing approach to food” proposal has further enriched and expanded the connotation and extension of food production and consumption. On the one hand, it is essential to re-understand and redefine food safety in the context of the “all-encompassing approach to food”; on the other hand, it is necessary to carefully consider the new requirements for agricultural production under the “all-encompassing approach to food”. The relationship between food production and consumption warrants particular attention, especially from a diversification perspective.

At the micro--level, low food consumption (diet) diversity remains a significant problem plaguing smallholder farmers, especially farmers in less--developed regions, and an important aspect influencing their welfare improvement [5]. Therefore, it is crucial to improve smallholder farmers’ nutritional health [6,7]. Given the ‘subsistence’ character of smallholder production, agricultural production diversification is often seen as an effective strategy for smallholders to improve their diversity and nutritional status [8,9]. 

Currently, evidence on the effectiveness of this strategy needs to be more comprehensive. Extant studies have shown that agricultural production diversity is considered the foremost determinant of dietary quality [10], with a positive effect on dietary diversity [8,11,12,13,14,15,16,17,18,19,20,21,22,23,24,25,26]. However, some scholars have raised questions about the effectiveness of enhancing dietary diversity through agricultural production diversity [27,28,29,30,31,32], contending that it does not significantly influence dietary diversity [33]. Thus, the results of the extant literature are heterogeneous [34] and have yet to reflect the trends in the research field, with insufficient attention devoted to the extent and depth of the research.

Based on the above gaps, this paper takes smallholder farmers as the research object, diversity as the keyword, and food security as the entry point and comprehensively analyzes the existing studies on the relationship between agricultural production diversity and dietary diversity/nutrition, intending to answer the following questions: (i) Comprehensively analyzing the publication trends, journals, authors, institutions, and countries; aiming at identifying the aggregated clusters shaping the research field; and exploring the literature and collaborators deserving analysis; (ii) Based on the bibliometric data, an in-depth insight into the research mainstream and research content from the existing literature is provided to identify future research directions further. It provides a reference for smallholder farmers to implement agricultural production diversification strategies and a comprehensive response guide for scientists, researchers, and professionals in production choices and food security. 

## 2. Methodology

Bibliometric citation meta-analysis is a primary format for analyzing literature targeting a given research topic. This paper adopts an econometric citation analysis method to quantitatively analyze the publication situation, authors, institutions, journals, and countries regarding this research topic based on the existing relevant literature, aiming to help readers identify the association between farmers’ production choices and food security, assess the current research trends in this field, recognize future research directions, and provide references towards realizing the policy of agricultural production and food security polices at the micro-level. 

Following this paper’s research topic and purpose, the authors conducted a literature search based on the “Web of Science” database, with keywords, abstracts, and titles as search criteria. The search items for production diversity were “crop diver*”, “plant diver*”, “production diver*”, and “biodiver*”; and the search terms for dietary diversity were “diet diver*” and “food security”. The research literature informed the selection criteria, focusing solely on studies that directly examined the relationship between agricultural production diversity and dietary diversity as their primary focus. Studies that merely mentioned production diversity and dietary diversity in passing in the introduction or discussion sections, even if they touched upon other aspects related to production diversity or dietary diversity, were excluded. According to this selection criterion, 46 articles corresponding to the research criteria were obtained (see Figure 1).

Based on the screening of the literature, we analyzed the bibliometric data in the bibliometrix software package and calculated output and impact metrics, also categorizing journals, authors, articles, and institutions. Then, we visualized the relevant data using VOS Viewer 1.6.19 (an econometrics web software tool for constructing visualizations of the literature) to understand co-citation links between published articles. In addition, we conducted a thorough review of the article content to help identify research trends and the current status.

## 3. Results and Discussion

The results of the data analysis displayed the multidisciplinary nature of the research field. From the study, we found that over 80% of the 46 articles were categorized in disciplines including Food Science Technology (19.3%), Business Economics (15.2%), Nutrition Dietetics (15.2%), Agriculture (14.5%), Plant Sciences (11.0%), and Science Technology Other Topics (4.8%).

### 3.1. Publication Trends

As shown in Figure 2, research levels on agricultural production diversity and dietary diversity have shown a rise in volatility since 2003. Overall, 98.3% of the 46 published articles were published in journals during the past ten years, with 14.6% of the articles published in 2023, ranking first in the last five years.

### 3.2. Most Influential Institutions

The number of publications and the total citations in agricultural production diversity and dietary diversity represent the significance of different research institutions. Our findings show that some institutional influence leads this research field. Table 1 presents the five most influential institutions involved in agricultural production diversity and dietary diversity, including the number of publications (left side of the table) and citations (right side of the table), contributing 52.1% of the articles in this field. The top five institutions in terms of publication numbers (P_APD&DD_) in this field are the Consultative Group for International Agricultural Research (CGIAR), International Food Policy Research Institute (IFPRI), University of Gottingen (UG), Bangladesh Agricultural University Bau (BAUB), and University of Michigan (UM). In terms of total citations, UG, CGIAR, IFPRI are in the top three, respectively, with UG in particular ranking first for the total citations (694) of the four articles published. Overall, these agencies should be viewed as “centers of excellence” in the process of transforming agri-food systems. Therefore, the results of this study contribute to targeted cooperation between researchers and relevant target institutions.

### 3.3. Most Influential Countries

Research in the field of agricultural production and dietary diversity mainly focuses on a few countries. Figure 3 provides the top five ranking countries based on articles published by authors affiliated with institutions in a particular country, with researchers from Germany and the United States of America contributing over a third (37.5%) of the total research output in the field. In addition, research from the UK, Bangladesh, and India is also increasing. Notably, more relevant research is needed in China. Therefore, the results of the country rankings reflect the future direction of research in a given country and the lack of it at this stage.

### 3.4. Most Influential Journals

Using the existing literature, bibliometric citation analysis was employed to evaluate the impact and performance of journals in the field [35,36] and identify key journals that have significantly influenced agricultural production diversity and dietary diversity research. Table 2 presents the top five journals based on the total number of articles related to agricultural production diversity and dietary diversity and their impact measured by average citations per year. These journals, including Food Security, PloS One, Agriculture and Food Security, Food Policy, and Agricultural Economics Research Review, collectively account for 41% of all publications in this area, each having published at least three articles over the past two decades. Generally, food and agriculture journals are the most influential.

Figure 4 shows the overview of influential journals using a quadrant diagram. For further analysis of the results, the top five journals measured by publications were analyzed in greater detail, using the number of articles published and average citations per year as comprehensive indicators for assessing research in agricultural production diversity and dietary diversity. Specifically, the x-axis denotes the number of articles published in the journal, whereas the y-axis signifies the average citations received per year. We established four primary groups of journal classifications based on the mean values of both variables (P_APD&DD_ = 4 and TC/t = 7.84), quadrant A: high output and high citations; quadrant B: high output and low citations; quadrant C: low output and high citations; and quadrant D: low output and low citations. 

FS is situated in quadrant A, indicating high performance both quantitatively and qualitatively. On the other hand, FP is placed in quadrant C, demonstrating high citations and room for growth in terms of publication quantity. In contrast, AFS, AERR, and PO are located in quadrant D, requiring further improvement in both quality and quantity. The results of this study can be used by researchers to spread their research in the field of agricultural production and food safety.

### 3.5. Most Influential Authors

In recent years, a total of 156 researchers have made significant contributions to the research field of promoting agricultural production transformation and ensuring food security. However, most researchers (12.2%) have not received a single citation in their publications, resulting in an inadequate impact. The results of the study showed that 42 researchers received over 30 publication citations, which we defined as influential authors. In particular, Table 3 presents the top five most influential authors based on publication quantity (left side) and citation frequency (right side), respectively. From Table 3, two researchers’ ranking lists are different, with KT Sibhatu and M Qaim performing well in the field both quantitatively and qualitatively. This data analysis helps researchers to target potential collaborators in the field and to access the influential literature on the research fields of agricultural production diversity and dietary diversity.

### 3.6. Research Themes

The co-occurrence of the keywords analyzed shows the mainstream research in the field as the keywords represent the research content in the articles. The study results showed 1425 keywords, with 50 occurring over six times. The evaluation criteria include the frequency of occurrences, links (representing co-occurrences with other keywords), and total link strength (indicating the number of publications where both keywords appeared together). The top ten keywords are presented in Table 4, with a total link strength of 46 for all keywords, indicating that at least two of the above keywords appear in the research literature.

We constructed a visual keyword co-occurrence graph using VOS Viewer. Figure 5 shows three clusters, with different colors indicating each cluster in the figure. The first cluster contains twenty items, the second cluster contains eighteen items, and the last cluster contains twelve keywords. Specifically, the third cluster contains the essential keyword “diet diversity” and the diet status of particular populations, such as women, households, and children. In the first cluster, our study also found keywords related to the country name, such as Bangladesh, country, etc., reflecting the study’s specific origin. The second group of keywords shows the relationship with dietary diversity in the context of agricultural production diversity (association, relationship, and analysis) and the pathways to improve dietary quality (market, market access, and income).

While the keyword analysis enabled us to provide readers with a general overview of the content of published research in agricultural production and food security, we also conducted an in-depth analysis of the content of 46 articles, aiming at identifying the central structure and specific content of studies on agricultural production diversity and dietary diversity (as shown in Table 5).

#### 3.6.1. Research Findings

The existing literature suggests that agricultural production diversity has the potential to influence households’ dietary diversity, with mixed results [57]. 

Extant studies have shown that agricultural production diversity is the most important determinant of dietary quality [10], with a positive effect on dietary diversity [8,11,12,13,14,15,16,17,18,19,20,21,22,23,24,25,26]. For instance, Singh et al. (2020) analyzed the impact of farm diversification on diet diversity in India [44]. That study found that crop diversity leads to an increase in both the quality and quantity of household diets, and the same evidence was found in another study in Bangladesh [45,47]. Studies by Ekesa et al. (2009) and Verma et al. (2007) both noted that agricultural biodiversity contributes to household dietary diversity [55,56]. 

Some scholars have questioned the strategy of improving dietary diversity through agricultural production diversity [27,28,29,30,31,32], arguing that agricultural production diversity is not a primary factor influencing dietary diversity [33,34]. For instance, Rosenberg et al. (2018) used survey data in Zambia to investigate the relationship between agricultural diversity and dietary diversity, and they found that agricultural production diversity had no significant effect on dietary quality [49]. Similarly, Cleghorn (2014) found no correlation between agrobiodiversity and dietary diversity [27]. In addition, Sibhatu et al. (2015) argued that market access has positive effects on dietary diversity, which are larger than those of increased production diversity [34]. 

However, some researchers have also stated that agricultural production diversification is detrimental to improving the quality of household meals [46,51]. For instance, Hlatshwayo et al. (2023) examined the relationship between agricultural production diversity, food security, and nutrition status using national survey data from South Africa. They found that increased diversified agricultural production was negatively associated with increased household dietary diversity [40]. Another study in Ghana revealed that crop diversity measured by the Simpson index (SI) was negatively associated with the dietary diversity of children [53].

#### 3.6.2. Impact Pathways

The underlying causal mechanisms still need to be better understood between agricultural production diversity and dietary diversity. Considering that small farmers are micro-operators integrating production and consumption, the influential mechanism of agricultural production on food security deserves deeper exploration (Figure 6).

Targeting agricultural production directly in terms of consumption is an essential way to improve the diet quality. In rural areas, smallholder farmers remain heavily dependent on their agricultural production activities for food consumption due to their inherent production, market, and capital vulnerability [45,58]. Agricultural production directly affects household dietary quality, i.e., diversified agricultural production will likely improve household meal levels [41,59]. Kim et al. (2018) found an example of support in a tracking survey about infant and young child feeding in Bangladesh, which found that children in livestock-keeping households were five times more likely to consume animal-based foods than other households [60].

Utilizing market access is a novel approach to amplify the influence of agricultural production diversification on food security. During the transformation from a smallholder productive operational system to a moderate-scale operation, production specialization weakens the direct effect of agricultural production diversity on dietary diversity. Thus, market participation becomes an alternative pathway to increase dietary diversity [34]. Generally, smallholder farmers allocate agricultural income generated from the sale of farm products to household production and consumption. Specifically, increased expenditure on food consumption contributes to improved dietary quality [56], while purchasing agricultural inputs such as seeds and fertilizers contributes to agricultural production diversification. For example, Esaryk et al. (2021), exploring the relationship between agricultural production diversity and dietary diversity among schoolchildren in Vietnam and Ethiopia, found that market access positively moderated the relationship, which is in line with the findings of another study in Bangladesh [38].

In general, diversification of agricultural production is the primary way to improve food and nutritional security among smallholder families with low socio-economic status, inaccessible transportation, and poverty. Esaryk et al. (2021) identified a significant association between crop diversity and children’s dietary diversity scores, particularly notable in impoverished households in Ethiopia and subsistence-oriented households in Vietnam [43]. However, market access and trade have more potential to improve dietary diversity among smallholder households with well-developed markets and higher income levels [34]. Rajendran et al. (2017) found that revenue generated from diversified crops is much more critical and significant to dietary diversity in Tanzania [33]. Similarly, Sibhatu et al. (2018) argued that diverse subsistence production tends to have a lesser impact on dietary diversity compared to the cash income derived from market sales [28].

#### 3.6.3. Approach to Measurement

Differences in measurement approaches lead to heterogeneity in findings, as shown in some studies [29,46,51,52,53]. Thus, a comprehensive analysis of agricultural production and dietary diversity measuring approaches is essential for the subsequent research. 

Based on the existing literature, the significant measures of agricultural production diversity include:Crop Counts. The number of livestock species was added to the number of crop species as measured over one year, i.e., for each species of crop planted or livestock raised, one point was assigned accordingly [8,11,13,16,18,19,22,25,26,28,30,32,34,40,41,42,43,47,49,52,53].Food Group Production Diversity Score (FGPD). Crops grown and livestock raised by farmers were specifically categorized into nine major food groups: cereals, pulses, stems and tubers, vegetables, fruits, meat, fish, eggs, and milk, measured over one year, with the production diversity score increasing by one point for each added food group [12,14,15,17,23,28,34,37,42].Simpson Index (SI). The SI was initially applied in ecological studies to assess biodiversity levels and later in agricultural production studies to measure the enrichment and evenness of crop species [61]. The calculation method is as follows:
(1)$$SI=1-\sum_{i=1}^{i}P_{i}^{2}$$
where $P_{i}$ denotes the proportion of the cultivated area of crop *i* to the total cultivated area of the crop. The SI takes a value between 0 and 1, with 0 indicating a single crop and 1 showing an evenly distributed cultivated area of all the crops [10,19,24,30,33,38,42,44,45,52,53].

Shannon–Wiener Diversity Index (SWDI). The SWDI is primarily used to measure the richness and evenness of crop species. The formula is as follows:(2)$$H=-\sum P_{i}{lnP}_{i}$$E = H/Hmax, Hmax = ln(S)(3)
where $P_{i}$ denotes the proportion of crop *i* to the total number of crops, S represents the total number of crops, and E means the evenness. The SWDI (H) value may be greater than 1, and evenness (E) ranges between 0 and 1. In actual calculations, the higher the diversity level, the lower the evenness [14,15,21,24,27].

Based on the existing literature, the significant measures of dietary diversity include:Household Dietary Diversity Score (HDDS). The indicator is based on 24 h [14,23,27,29,30,37,39,40,47,50,51,52] or seven-day recall data [10,12,15,17,19,25,29,55] by weighting the consumption of the 12 major food groups (cereals, stems and tubers, vegetables, fruits, meat, eggs, fish and other aquatic products, milk, legumes and nuts, fats and oils, sugars, and condiments). The calculation formula is as follows:
(4)$$HDDS=\sum_{i=1}^{12}n$$

In the specific calculations, the consumption frequency was calculated based on whether the food group was consumed in the past period, i.e., one point was assigned to consumption and vice versa.

Individual Dietary Diversity Score (IDDS). The IDDS is calculated similarly to the HDDS [49,51], with the difference of re-categorizing the food groups according to populations such as women and children. Specifically, the Women’s Dietary Diversity Score (WDDS), a specialized indicator for evaluating women’s dietary diversity, classifies food groups into cereals, legumes, nuts, dairy, meat, eggs, vitamin A-rich green leafy vegetables, other vitamin A-rich vegetables and fruits, other vegetables, and other fruits. Higher scores on the WDDS indicate a higher quality diet for women [13,19,32,42,47]. The CDDS is an indicator specifically designed to measure children’s dietary quality, only considering the seven major food groups, i.e., grains, stems, and tubers; legumes and nuts; dairy; meat; eggs; vitamin A-rich vegetables and fruits; and other vegetables and fruits. Thus, children’s dietary diversity scores range from 0 to 7 [13,18,19,32,43,44,53].

## 4. Conclusions

Currently, relatively extensive research on production strategies and food security focuses mainly on smallholder farmers at the micro-level in developing countries. This paper provides a roadmap for agricultural production and food security researchers by conducting a systematic review of the literature, summarizing some research methods and perspectives applicable to local socio-economic development. Based on the extant literature, the following conclusions are presented: (i) Research trends on agricultural production diversity and dietary diversity have shown a rise in volatility since 2003; (ii) Our findings show that some institutional influence leads this research field, such as UG, CGIAR, and IFPRI; (iii) Research in the field of agricultural production diversity and dietary diversity has mainly focused on a few countries, such as Germany, the United States, and the United Kingdom; (iv) Food and agriculture journals, such as Food Security, PloS One, Agriculture and Food Security, Food Policy, and Agricultural Economics Research Review, are the most influential; (v) The two most influential authors in this research field are KT Sibhatu and M Qaim; (vi) The co-occurrence of the keywords analyzed shows the mainstream research as the keywords represent the research content in the articles.

While the keyword analysis enabled us to provide readers with a general overview of the content of published research in agricultural production and food security, we also conducted an in-depth analysis of the content of 46 articles, aiming at identifying the central structure and specific content of studies on agricultural production diversity and dietary diversity. The study shows that agricultural production diversity has the potential to influence households’ dietary diversity, with mixed results [57,60]. The underlying causal mechanisms still need to be better understood between agricultural production diversity and dietary diversity. Specifically, targeting agricultural production directly in terms of consumption is an essential way to improve the diet quality. Market access is a novel pathway to enhance the impact of agricultural production diversification on food security. In general, diversification of agricultural production is the primary way to improve food and nutritional security among smallholder families with low socio-economic status, inaccessible transportation, and poverty. However, market access and trade have more potential to improve dietary diversity among smallholder households with well-developed markets and higher income levels [34]. Differences in measurement approaches lead to heterogeneity in findings. Based on the existing literature, the significant measures of agricultural production diversity include Crop Counts, FGPD, SI, and SWDI; the significant measures of dietary diversity include HDDS and IDDS. 

This paper provides a comprehensive overview and analysis of the correlation between agricultural production and food security. The implications of the study’s findings centered around how smallholder farmers achieve interaction and coordination between agricultural production and food and nutrition security, as well as ensuring household food security through implementing agricultural production diversification strategies or enhanced market participation. Therefore, in less-developed regions of China, improving the well-being of smallholder farmers is critical to restructuring a nutritious and healthy diet, more precisely, by shifting to diversified food consumption. This paper provides a systematic analysis of agricultural production and dietary diversity, contributing to researchers, policymakers, and professionals’ knowledge on the differentiated approaches to address the severe challenges of diversity and nutritional status of various smallholder farmers, and has realistic value for the transformation of agri-food systems.

Even though existing studies provide much work in related areas, they have several limitations. On the one hand, most existing studies used cross-sectional data to measure the correlation between agricultural production and food security, and only a few studies used panel data [12,13,21,23,43,47]. Therefore, the study results should not be over-interpreted in a causal significance. On the other hand, the collected data for studies involving household diets are affected by seasonality, which could lead to biased findings [48].

This paper explores the correlation between agricultural production choices and food security based on the perspective of micro-diversity. It puts forward several insights into the future development of small farmers in production, diet, and other aspects: Firstly, we must comprehensively and profoundly understand the connotation of the transformation of the agri-food system to reasonably adjust the dietary structure and gradually realize the transformation from cereal-based consumption to the diversification of meat, vegetables, fruits, aquatic products, etc.; Secondly, it is necessary to strengthen market integration, promote market participation by small farmers, and broaden income-generating channels; Lastly, we must explore innovative agricultural technologies (such as genetically modified varieties) suitable for small farmers’ development, providing technological guarantees for production diversification in rural areas and enhancing agricultural production and food security development.

## Figures and Tables

**Figure 1 foods-13-00771-f001:**
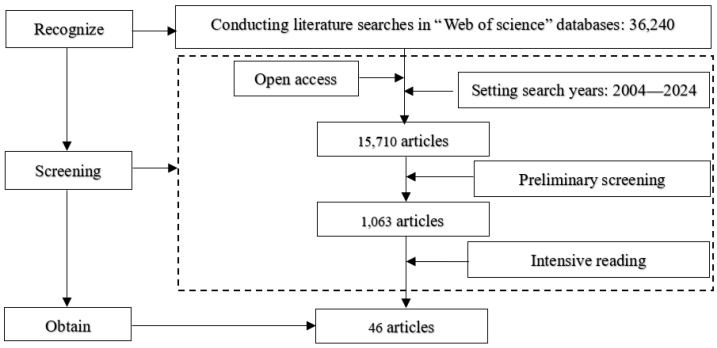
Roadmap for literature search.

**Figure 2 foods-13-00771-f002:**
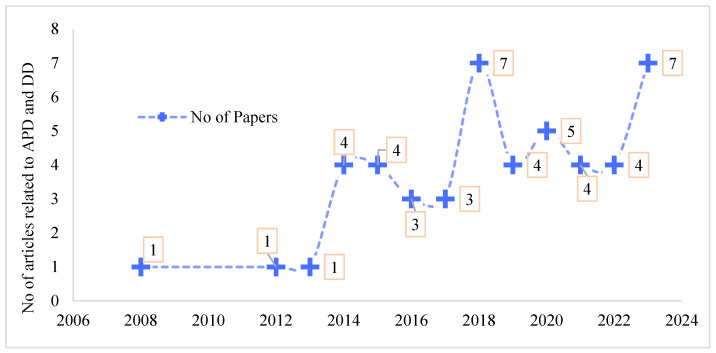
Publication trend related to agricultural production diversity and diet diversity. Note: APD denotes agricultural production diversity; DD denotes dietary diversity.

**Figure 3 foods-13-00771-f003:**
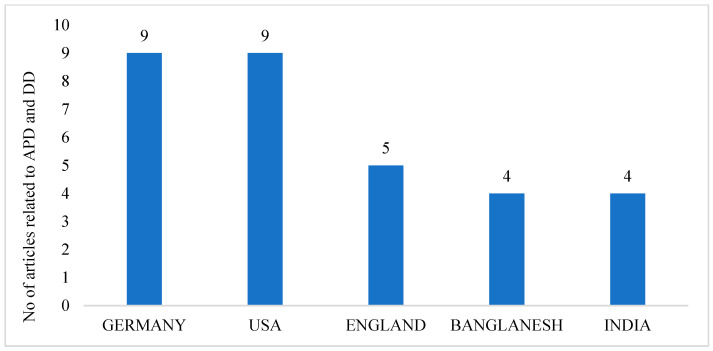
Top 5 countries regarding the number of published articles.

**Figure 4 foods-13-00771-f004:**
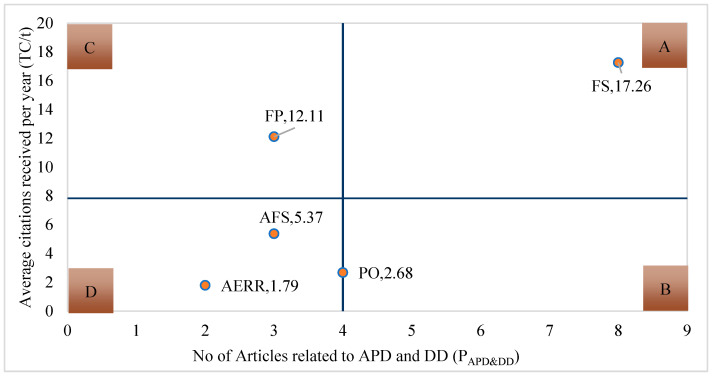
Journal’s impact on agricultural production diversity and diet diversity studies.

**Figure 5 foods-13-00771-f005:**
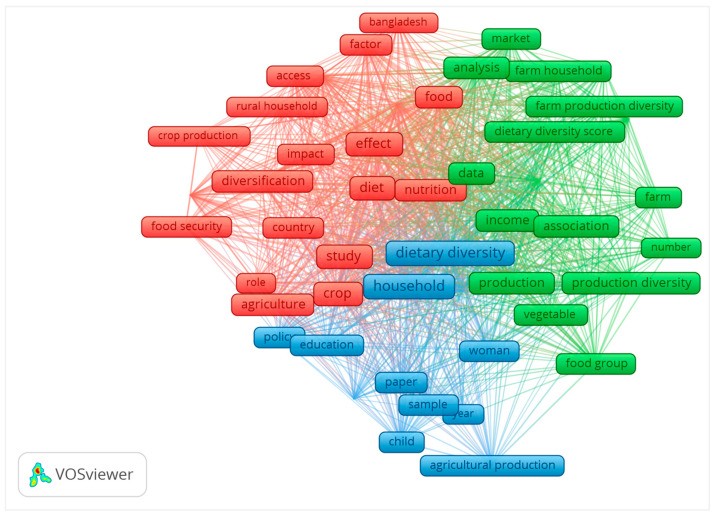
Co-occurrence map of keywords.

**Figure 6 foods-13-00771-f006:**
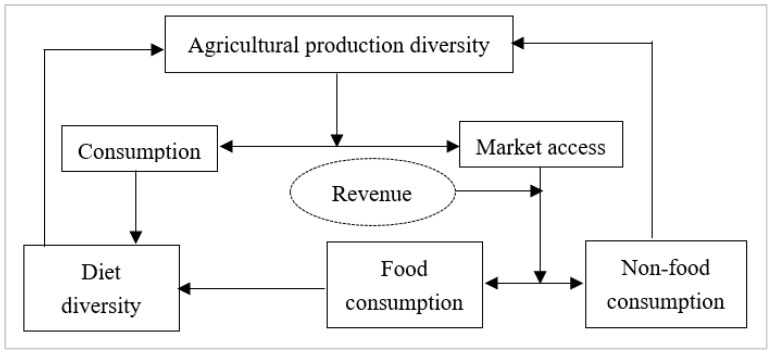
Impact path diagram.

**Table 1 foods-13-00771-t001:** Most influential institutions.

Institutions	P_APD&DD_	TC	TC/P
CGIAR	9	217	24.11
IFPRI	6	106	17.67
UG	4	694	173.5
BAUB	3	46	15.33
UM	3	47	15.67

Note: P_APD&DD_ indicates the number of publications related to the field of agricultural production and dietary diversity; TC represents the number of times an article was cited by other publications in the Web of Science database, i.e., total citations received; TC/P indicates citations per published article.

**Table 2 foods-13-00771-t002:** Most influential journals.

Rank	Journal	Label	P_APD&DD_	TC	TC/P	H-Index	TC/t
1	Food Security	FS	8	328	41	7	17.26
2	PloS One	PO	4	51	12.75	3	2.68
3	Agriculture and Food Security	AFS	3	102	34	3	5.37
4	Food Policy	FP	3	230	76.67	2	12.11
5	Agricultural Economics Research Review	AERR	2	34	17	2	1.79

Note: Ranking sorted by P_APD&DD_.

**Table 3 foods-13-00771-t003:** Most influential authors.

Rank	Author	P_APD&DD_	TC	TC/P	Rank	Author	P_APD&DD_	TC	TC/P
1	Jones A D	3	47	15.67	1	Sibhatu K T	3	682	227.33
2	Qaim M	3	620	206.67	2	Qaim M	3	620	206.67
3	Sibhatu K T	3	682	227.33	3	Krishna V V	1	367	367
4	Alam M J	2	0	/	4	Pellegrini L	1	127	127
5	Begum I A	2	0	/	5	Tasciotti L	1	127	127

Note: Ranking sorted by P_APD&DD_ (left); ranking sorted by TC (right).

**Table 4 foods-13-00771-t004:** Important keywords.

Rank	Keyword	Occurrences	Total Link Strength
1	Dietary diversity	39	46
2	Household	35	46
3	Effect	29	46
4	Diet	26	46
5	Crop	25	46
6	Study	24	46
7	Production	22	46
8	Data	22	46
9	Diversity	22	46
10	Food	21	46

**Table 5 foods-13-00771-t005:** Overview of articles included in methods, main findings, and conclusions.

Original Study Reference	County	Sample Size	Data Type	Measures of Dietary Diversity	Measures of Agricultural Production Diversity	Summary Findings
Tacconi et al. (2023) [37]	South and Southeast Asia	4772 HH	HS	HDDS (24 h recall)	FGPD	Environmental and climate variables significantly influence farm diversification. Positive association between farm diversification, market orientation, off-farm income generation, HDDS, and WDDS. The impact of farm diversification on HDDS was found to be more pronounced among smaller farms, while diminishing on larger farms.
Chanmi et al. (2023) [38]	Cambodia	6736 HH	HS	FCS	SI	Farm production diversification can positively affect the diversity of household diets. Market access moderating the effect of agricultural production diversification on household dietary diversity.
Fatch et al. (2023) [39]	Malawi	424 HH	HS	HDDS (24 h recall)	PCA	Positive association between agricultural diversity and dietary diversity (WDDS, CDDS, and MDDS); No significant association between agricultural diversity and the nutritional status of children, women, and men.
Hlatshwayo et al. (2023) [40]	South Africa	1520 HH	NR	HDDS (24 h recall) HFIAS	CC	Ownership of livestock, harvest, and disability within the family had a negative impact on the food security status of smallholder farmers. Household size has a positive effect on food security and nutrition status. Social grants, agricultural assistance, and harvest have a detrimental impact on nutritional status.
Ahmadzai et al. (2023) [11]	Afghanistan	28,071 HH	NR	HFCS	CC	There is a positive relationship between crop diversification and household consumption as well as dietary diversity.
Mastura et al. (2023) [12]	Bangladesh	6503 HH	HS	HDDS (day recall) HFVS	FGPD; PDS	Positive association between agricultural production diversification and household dietary diversity. Negative association between market access and household dietary diversity. Positive association between participation in non-farming activities and dietary diversity.
Alam et al. (2023) [13]	Bangladesh	6503 HH	HS	MDDS, WDDS, CDDS (24 h recall)	CC	The diversification of agricultural production can promote dietary variety across all life stages of farm household members. Agricultural production diversification significantly and positively influences Minimum Dietary Diversity Score (MDDS), Women’s Dietary Diversity Score (WDDS), and Child Dietary Diversity Score (CDDS).
Khandoker et al. (2022) [14]	Bangladesh	11,720 HH	HS	HDDS (24 h recall) FVS	SWDI; FGPD	Positive associations between farm production diversity, production diversity, income, and dietary diversity.
Kuntashula et al. (2022) [29]	Zambia	7934 HH	NR	HDDS (24 h recall)	SI; CC; AID; HID	Positive association between Farmer Input Support Policy (FISP) and production diversity. Positive association between FISP and household dietary diversity. Mixed results between FISP and monthly food inadequacy. No significant association between production diversity and household dietary diversity.
Bernzen et al. (2022) [41]	Bangladesh	1188 HH	HS	FCS	CC	Households cultivating shrimp have a significantly higher dietary diversity than households that do not. Positive association between crop diversification and shrimp farmers’ dietary diversity.
Lourme-Ruiz et al. (2021) [42]	Burkina Faso	579 HH	HS	WDDS (24 h recall)	CC; SI; FGPD; NFD	WDDS is positively associated with PDS and the number of agroforestry tree species. WDDS is negatively associated with cotton production when managed by male farm heads but positively when managed by women.
Esaryk et al. (2021) [43]	Vietnam Ethiopia	1012 HH (Ethiopia) 1083 HH (Vietnam)	HS	CDDS (24 h recall)	CSR; CNFR	There is a small, positive association between household CNFR and CDDS in Ethiopia but not in Vietnam. The correlation between crop diversity and Child Dietary Diversity Score (CDDS) is most pronounced in impoverished households in Ethiopia and subsistence-oriented households in Vietnam. Agricultural earnings positively mediated the association between crop diversity and diet diversity in Ethiopia.
Azzarri et al. (2021) [15]	Malawi	935 HH	HS	HDDS (7-day recall)	SWDI; FGPD	Production diversity has a significant positive treatment effect on dietary diversity. There is a positive correlation between agricultural income and dietary diversity.
Obisesan et al. (2021) [16]	Nigeria	1226 HH	HS	HDDS (7-day recall)	CC	Crop diversification positively (*p* < 0.01) affects households’ dietary diversity.
Mehraban et al. (2021) [17]	Indonesian	2785 HH	HS	HDDS (7-day recall)	FGPD (PD9, PD10, and PD12)	Positive relationships between production diversity and household dietary diversity, as well as between market access and household dietary diversity. Negative relationship between specialization and dietary quality during 2000–2015.
Gupta et al. (2020) [18]	India	3600 HH	HS	HDDS, CDDS (7-day recall; 24 h recall)	CC	Positive relationship between food groups grown and diet diversity. On-farm production is significantly associated with improved dietary diversity scores.
Muthini et al. (2020) [19]	Kenya	779 HH	HS	HDDS (7-day recall) CDDS, MDDS (24 h recall)	SI; CC	Farm production diversity is significantly associated with the dietary diversity of women and households. No association between farm production diversity and the dietary diversity of children.
Argaw et al. (2020) [10]	Ethiopia	1236 FHH 5185 MHH	RS	FVS, HDDS (7-day recall)	SI	Positive relationships between gender, production diversity, and dietary diversity. Production diversity is the primary driver of dietary diversity. In terms of gender-based disparities in dietary diversity, the source of food—whether it originates from within the household’s production system or is acquired from the market—plays a crucial role.
Mulwa et al. (2020) [20]	Namibia	650 HH	HS	DDS (24 h recall)	HI	Positive relationships between crop production diversification, livestock production diversification, and food security outcomes.
Singh et al. (2020) [44]	India	1106 HH	HS	HDDS, WDDS, CDDS (24 h recall)	SI	Crop diversity has a positive association with dietary diversity among adults (both men and women); Positive relationship between crop diversity and dietary diversity among adolescents and children in Haryana.
Baye et al. (2019) [21]	Ethiopia	40,000 HH	NR	NI	SWDI	Positive relationship between diets and diversifying production.
Uddin (2019) [30]	Bangladesh	6503 HH	NR	HDDS, DDI (24 h recall)	RI; SI; EI	There is no notable correlation between crop diversification and dietary diversity.
Pinto et al. (2019) [45]	Bangladesh	6500 HH	HS	HAZ, WHZ, WAZ	GSI	Positive association between women’s empowerment and crop diversity. Positive association between diversification and reducing climate risk and children’s nutritional status.
Mango et al. (2018) [46]	Malawi	271 HH	HS	FCS, HFIAS	CDI	Positive association between crop diversification and household FCS. Negative association between crop diversification and HFIAS.
Islam et al. (2018) [47]	Bangladesh	6715 HH 6071 WO	HS	HDDS, MDDS, FVS	CC; MSRI	Positive association between farm diversification and diet diversity.
Luna-González et al. (2018) [31]	Guatemala	154 CH 127 MH	HS	CDDS (24 h recall)	CC; AC	Positive relationships between food self-sufficiency; nutritional, functional diversity, dietary diversity scores; and crop and animal species richness. The local market was negatively correlated with dietary diversity scores. No relationship between dietary diversity and child anthropometric status. Positive relationships between child anthropometric status, sanitary conditions, and maternal education. Negative relationship between household size and frequent child morbidity.
Sibhatu et al. (2018) [28]	26 countries	45 RA	LR	HDDS (different recall periods)	Different diversity methods	Positive correlations exist among food self-sufficiency, nutritional diversity, functional diversity, dietary diversity scores, as well as crop and animal species richness. No significant association between farm production diversity and smallholder diets and nutrition in around 40% of studies.
Sibhatu et al. (2018) [48]	Indonesia, Kenya, and Uganda	672 HH (Indonesia) 393 HH (Kenya) 419 HH (Uganda)	HS	HDDS (7-day recall)	CC; FGPD	Production diversity measured by simple species count is positively associated with most dietary indicators. Production diversity measured by the number of food groups produced is insignificant and associated with dietary diversity. Diverse subsistence production often contributes less to dietary diversity than cash income from market sales.
Somé et al. (2018) [32]	Burkina Faso	10,790 HH	RS	WDDS (7-day recall)	CC	Crop diversity cannot modify the seasonal differences in the diversity of household diets.
Rosenberg et al. (2018) [49]	Zambia	131 SEAs	HS	IDDS (24 h recall) HHS, HFIAS	CC	No significant association between agricultural diversity and dietary diversity (children or their mothers).
Kennedy et al. (2017) [50]	Low-income countries	Multiple	RS	HDDS (24 h recall)	SDI	Production diversity is associated with improved diet diversity in most cases. Women play a pivotal role in two key pathways: (1) through the consumption of products derived from their own production or gathered from the wild; (2) through the purchase of biodiversity, whether wild or cultivated.
Gondwe et al. (2017) [22]	Zambia	542 HH	HS	DDS (24 h recall)	CC	Positive relationship between crop production and household diets.
Rajendran et al. (2017) [33]	Tanzania	300 HH	HS	DDS (24 h recall) FVS	SI	NO association between Simpson’s index and dietary diversity. Revenue generated from diversified crops is much more critical and significant to dietary diversity.
Chinnadurai et al. (2016) [51]	Tamil Nadu	45 CR	RS	HDDS, IDDS (24 h recall)	HI	Positive association between crop diversification and dietary diversification. Negative association between vegetable diversification and diet diversification.
Kavitha et al. (2016) [52]	Telangana, Maharashtra	289 HH	HS	HDDS (24 h recall)	CC; SI	Positive association between crop diversity and household dietary diversity at the bivariate level. No significant association between crop diversity and dietary diversity in the multiple linear regression model.
Argyropoulou et al. (2016) [53]	Ghana	329 HH	CS	CDDS (24 h recall)	CC; SI	No significant association between crop diversity and CDDS. No significant association between crop diversity and the nutritional status of children. SI is negatively associated with CDDS.
Sibhatu et al. (2015) [34]	Indonesia, Kenya, Ethiopia, and Malawi	8230 HH	NR	DDS (24 h recall) FVS	CC; FGPD	In certain scenarios, production diversity shows a positive association with dietary diversity, though this relationship is not universal. Positive association between production diversity and dietary diversity turns to not significant or even negative because of foregone income benefits from specialization. The positive impact of market access on dietary diversity surpasses that of increased production diversity.
Dillon et al. (2015) [23]	Nigeria	5000 HH	GHS-Panel	HDDS (24 h recall)	FGPD	Positive associations between crop diversity, agricultural revenue, and dietary diversity (CDDS).
Romeo et al. (2015) [24]	Kenya	1353 HH	HS	HDDS (7-day recall)	SI; SWDI	Positive association between agricultural production practices and household food diversification.
Snapp et al. (2015) [25]	Malawi	9189 HH	HS	HDDS, FCS (7-day recall)	CC	Positive association between crop diversity and dietary diversity. Positive association between education, income, market access, and dietary diversity.
Cleghorn (2014) [27]	Tanzania	122 HH	HS	HDDS (24 h recall)	SWDI	No significant association between dietary diversity and nutritional status. Negative association between dietary diversity and height for age z-scores. No significant association between agrobiodiversity and dietary diversity.
Pellegrini et al. (2014) [26]	Eight developing Countries ^a^	800 HH	NR	Food (group) count	CC	Positive correlation between the number of crops cultivated, household income from crops, and dietary diversity.
Fanzo et al. (2013) [8]	Nepal Bangladesh	741 MC	CS	Food count	CC	Positive correlation between agricultural production diversity and food security, dietary intake, and nutritional status.
Goshu et al. (2012) [54]	Ethiopia	260 HH	HS	Food (group) count	CDI	No significant associations between crop diversification, food security, and dietary diversity status.
Ekesa et al. (2009) [55]	Kenya	144 HH	HS	HDDS (7-day recall)	CC; AC	Positive association between agricultural biodiversity and dietary diversity (CDDS).
Verma et al. (2007) [56]	India	Urban and rural people	RS	CP	CC	Positive association between non-cereal commodities and diversified production. Positive association between diversification and producers’ income. Positive association between diversified food basket and food security.

Note: Articles sorted by year of publication; HH, households; HS, household surveys; HDDS, household dietary diversity score; FGPD, food group production diversity score; FCS, food consumption score; SI, Simpson index; PCA, principal component analysis; NR, national representative sample; HFIAS, household food insecurity access scale; CC, crop count; HFCS, household food consumption score; HFVS, household food variety score; PDS, production diversification score; MDDS, men dietary diversity score; WDDS, women dietary diversity score; CDDS, children dietary diversity score; FVS, food variety score; SWDI, Shannon–Wiener diversity index; GHS-Panel, General Household Survey-Panel; AID, agricultural income diversity; HID, household income diversity; NFD, nutritional functional diversity; CSR, crop species richness; FHH, female-headed household; MHH, male-headed household; CNFR, crop nutritional functional richness; HI, Herfindahl index; DDI, Dietary Diversity Index; NI, nutrient intake; RI, Rice Share Index; EI, Entropy Index; HAZ, height-for-age z-scores; WHZ, weight-for-height z-scores; WAZ, weight-for-age z-scores; GSI, Gini–Simpson index; CDI, crop diversification index; WO, women; MSRI, Margalef species richness index; CH, children; MH, mother; AC, animal count; RA, research articles; LR, literature research; SEAs, a unit of area used for the Zambian census; HHS, household hunger scale; CR, crop; CS, case study; MC, mother–child; CP, consumption pattern. ^a^ Malawi, 2004; Nepal, 2003; Vietnam, 1998; Pakistan, 2001; Nicaragua, 2001; Indonesia, 2000; Albania, 2005; Panama, 2003.

## Data Availability

The original contributions presented in the study are included in the article, further inquiries can be directed to the corresponding author.

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
