# Peer review of "Production Choices and Food Security: A Review of Studies Based on a Micro-Diversity Perspective"

_foods, 2024, doi:10.3390/foods13050771_

Round 1

Reviewer 1 Report

Comments and Suggestions for Authors

This current review  paper the main question addressed by the research provides a roadmap for agricultural production and food security researchers by using a systematic review of the literature, summarizing some research methods and perspectives applicable to local socio-economic development. What parts do you consider original or relevant for the field IS THE AGRICULtural gaps in the field does the paper address mainly  Overview of articles included in methods, main findings, and conclusions. study identifies the most influential journals, authors, organizations and countries, and reveals research themes related to agricultural production and food security, this data analysis can help researchers to target potential collaborators in the field and to access the influential literature on the research fields of agricultural production.

Comments Paper is sound, with several remarks Impact and my comments on the tables and figures and good quality of the data covering path diagram, Individual Dietary Diversity Score (IDDS), Crop Counts, FGPD, SI, SWDI;  significant measures of dietary diversity include HDDS and IDDS, etc with good explanations diagrams and figures makes more easily for readers. conclusions are consistent with the  evidence and arguments presented

Please revise English language accordingly.

Comments on the Quality of English Language

some ameliorations is good

Author Response

Dear reviewer,

The details are shown in the PDF file.

Thank you and best regards.

Yours sincerely,

Yanfang Huang

Reviewer 2 Report

Comments and Suggestions for Authors

The manuscript is written clearly and seems to provide the basis for interesting and relevant recommendations for policy and practice. It would be helpful for the authors to flesh out those recommendations further and to be a bit more explicit abot where the reviewed scholarship leads regarding policy and practice.

Figure 5 is quite difficult to read. If possible, enlarging the dimensions should help. Otherwise it may be necessary to modify the color scheme to permit more clarity of the verbiage.

Comments on the Quality of English Language

Some minor punctuation and grammatical issues are apparent, and could be dealt with easily in the final stage of manuscirpt copyediting.

Author Response

(The authors gave the same response as above.)

Reviewer 3 Report

Comments and Suggestions for Authors

The authors present a review focused on the bibliometric study of scientific articles and quantitatively analysis of publications, authorships and trends inherent to the specific topic of production choices and food security.
After a thorough analysis of the submitted material, in this reviewer's opinion, a review structured this way does not fall within the scope of the journal. Therefore, such a study cannot be accepted for publication unless it is restructured by focusing reviewing aspects of the selected papers on scientific content rather than on bibliometric data.

Comments on the Quality of English Language

Minor editing of English language required

Author Response

(The authors gave the same response as above.)
